# Tuberculosis Death Epidemiology and Its Associated Risk Factors in Sabah, Malaysia

**DOI:** 10.3390/ijerph18189740

**Published:** 2021-09-16

**Authors:** Richard Avoi, Yau Chun Liaw

**Affiliations:** 1Department of Community and Family Medicine, Faculty of Medicine and Health Sciences, Universiti Malaysia Sabah, Kota Kinabalu 88400, Malaysia; 2TB/Leprosy Unit, Sabah State Health Department, Ministry of Health, Putrajaya 62590, Malaysia; liaw608@gmail.com

**Keywords:** tuberculosis, case fatality, mortality, Sabah, Malaysia

## Abstract

Tuberculosis (TB) is a leading killer from a single infectious agent globally. In 2019, Malaysia’s TB incidence rate was 92 per 100,000 population, and the TB mortality rate was estimated at 4 cases per 100,000 population per year. However, the state of Sabah had a higher burden of TB with a notification rate of 128 per 100,000 population and a TB case fatality rate of 8% compared to the national figure. This study aims to provide a comprehensive report on TB deaths epidemiology and its associated factors at a sub-national level. This nested case-control study used Sabah State Health Department TB surveillance data from the Malaysia national case-based TB registry (MyTB) between 2014 and 2018. Cases were defined as all-cause TB deaths that occurred before anti-TB treatment completion from the time of TB diagnosis. Controls were randomly selected from TB patients who completed anti-TB treatment. The TB mortality rate had increased significantly from 9.0/100,000 population in 2014 to 11.4/100,000 population in 2018. The majority of TB deaths occurred in the first two months of treatment. TB-related deaths were primarily due to advanced disease or disseminated TB, whereas non-TB-related deaths were primarily due to existing comorbidities. Many important independent risk factors for TB deaths were identified which are useful to address the increasing TB mortality rate.

## 1. Introduction

Tuberculosis (TB) is caused by the bacterium *Mycobacterium tuberculosis* and is a leading killer from a single infectious agent globally, ranks higher than HIV/AIDS, and is one of the top 10 causes of death worldwide [1]. In 2019, an estimated 10 million people fell ill with TB; of these, 1.2 million TB deaths were among human immunodeficiency virus (HIV)-negative people, and 208,000 deaths among HIV-positive people [1]. The World Health Organization (WHO) has set the target of a 35% reduction in TB deaths by 2020 compared with 2015 as one of the targets to end the TB epidemic [2]. However, the WHO reports [1] that the annual number of TB deaths is not falling fast enough to reach the 2020 milestone, whereby the cumulative reduction between 2015 and 2019 was only 14%.

TB incidence and TB mortality rate are essential indicators of TB burden. In 2019, the estimate for TB incidence rate in Malaysia was 92 cases per 100,000 population, and the TB mortality rate was estimated at 4 cases per 100,000 population per year [1]. However, Sabah state had a higher burden of TB with a notification rate of 128 cases per 100,000 population and a TB case fatality rate of 8% compared to the national figure [3].

The TB control program in Malaysia has been strengthened to be aligned with the WHO End TB Strategy. The strengthening of the program is important to ensure that the targets will be achieved by 2035. Under the strategy, one of the targets is to reduce deaths due to TB by 35% in the year 2020, 75% in 2025, 90% in 2030, and finally reach the target of reduction by 95% in the year 2035 compared with 2015 [2]. TB control strategies and activities need to be tailored to different settings or states in Malaysia. To come up with a comprehensive national strategic plan for TB, an in-depth analysis of the TB disease epidemiology for each state is crucial. 

Studies looking into factors associated with TB deaths have been carried out worldwide in high and low TB incidence countries, and have found important factors which include patients’ sociodemographic characteristics, TB disease characteristics, TB treatment characteristics, and underlying comorbidities [4,5]. However, the findings are different between countries and regions. In Malaysia, studies investigating specifically TB death and its associated factors are limited. A recent study [6] reporting factors associated with all-cause mortality among TB patients was conducted at the national level; however, similarities or differences between states were not explored. TB death situation in the state of Sabah will be explored in this study. Sabah is located in East Malaysia and is economically less developed compared to the states in peninsular Malaysia. Sabah also has a large number of immigrants from the Philippines and Indonesia living in overcrowded settlements. Therefore, this study aims to provide a comprehensive report on TB deaths epidemiology and its associated factors specifically for the state of Sabah. Sabah state contributes the highest TB burden, i.e., 19.7% of Malaysia’s total reported TB cases [3].

## 2. Methods

### 2.1. Study Population and Source of Data

This nested case-control study used Sabah State Health Department TB surveillance data taken from the Malaysia national case-based TB registry (MyTB) between 2014 and 2018. Cases were defined as all-cause TB deaths that occurred before anti-TB treatment completion from the time of TB diagnosis. Controls were randomly selected from TB patients registered between 2014 and 2018 who completed anti-TB treatment. Sociodemographic, comorbid, disease, and treatment characteristics of cases and controls were extracted from MyTB. Sabah population data were obtained from the Department of Statistics Malaysia.

### 2.2. Definitions

#### 2.2.1. Outcome Variables

The main outcome variable was all-cause TB death, which was categorized into two groups: TB-related death and non-TB-related death. TB death in this study was defined as all-cause mortality that occurs before completing anti-TB treatment from the date of TB diagnosis. All TB death cases in MyTB will be classified into TB-related or non-TB-related death by the state and district level TB death audit committee. The members of the committee are a Chest physician, Family Medicine Specialist, TB program manager, doctors, and health staff involved in the management of TB patient. In this study, TB-related death was defined as the underlying cause of death being due to TB directly, whereas non-TB-related death was defined as any underlying cause of death not directly due to TB as decided by the TB audit committee.

#### 2.2.2. Explanatory Variables

The independent variables that may be associated with TB death were assessed, including sociodemographic variables (age, gender, place of residence, nationality, education level, and income status), comorbid variables (smoking status, diabetes mellitus, and human immunodeficiency virus infection), disease characteristics (category of TB, type of case detection, BCG scar, TB anatomical site, initial sputum acid-fast bacilli status, chest radiography findings at diagnosis, multi-drug resistant TB status, TB meningitis, and TB military), and treatment characteristics (treatment regime during intensive phase, directly observed treatment achievement during intensive phase and treatment delay, and treatment delay).

The lesion on chest radiography (CXR) findings at diagnosis was categorized into three groups, i.e., minimal, moderate, and advanced. Based on the Malaysia TB clinical practice guidelines, minimal lesion is defined as having slight lesions with no cavity and lesions confined to small parts of one or both lungs but the total extent not exceeding the upper zone; moderate lesion is defined as having dense lesions not exceeding one-third of one lung or both lungs not exceeding the volume of one lung, or presence of cavity not exceeding 4 cm in diameter; advanced lesion defined as lesions are more extensive than the moderate lesion.

Directly observed therapy (DOT) was defined as anti-TB medication ingestion that was directly supervised by health staff or trained family members. Under the TB control program, TB patients will take their medication under the supervision of either the health staff or a trained family member. The DOT supervisor will sign the TB treatment booklet after observing the patient ingesting the correct dosage of anti-TB drugs. The information on the achievement of DOT supervision was obtained from the TB treatment booklet. DOT supervision is achieved during the intensive phase when 80% of the prescribed daily doses are taken and supervised. Treatment delay in days was calculated by subtracting the two dates (i.e., date of treatment started minus date of diagnosis). An interval of 14 days or longer between the two dates was defined as a delay in starting anti-TB treatment. 

### 2.3. Statistical Analysis

Data were expressed as count and percentage for categorical variables and the mean ± standard deviation (SD) or median with interquartile range (IQR) for continuous variables. The sociodemographic, comorbid, disease, and treatment characteristics between all-cause TB deaths (cases) and controls were compared using Pearson’s chi-square test or Fisher’s exact test and the independent-samples *t*-test where appropriate. 

All variables found to be significant (*p* < 0.05) from the univariable analysis were considered for inclusion in multivariable analysis. Multivariable binary logistic regression was used to assess the associations of selected independent variables with all-cause TB deaths. A simultaneous forced-entry method was employed in multivariable analysis. The model fit was assessed using Hosmer–Lemeshow goodness-of-fit test, where a *p*-value greater than 0.05 means that the model is a good fit. A similar approach was also used to identify risk factors for all-cause TB deaths before anti-TB treatment started compared to after anti-TB treatment started. Time to death in days was calculated by subtracting the two dates (i.e., date of death minus date of treatment started). Death before treatment is initiated is defined when the difference between the two dates is zero. 

Multinominal logistic regression analysis was used for comparison between cases of TB-related death and non-TB-related death with the controls to identify the factors associated with TB-related and non-TB-related deaths. Crude odds ratio (cOR) and adjusted odds ratio (aOR) with a 95% confidence interval (95%CI) and Wald *p*-value were reported in all the regression analysis results to show the strength of association between independent and outcome variables. All data preparation, calculations, and analyses were performed using Microsoft Excel and IBM Statistical Package for the Social Science (SPSS) software version 26. The level of significance was set at a *p*-value less than 0.05. 

### 2.4. Ethical Consideration

Ethical approval for this study was obtained from the Medical Ethics Committee of the University Sabah Malaysia on 25 September 2020 JKEtika 4/20 (8), and from the Medical Research and Ethics committee (MREC), Ministry of Health Malaysia on 14 September 2020 (approval code NMRR-20-1289-55461). The Sabah State Health Director granted permission to access the de-identified case-listing of TB patients from the MyTB database, and patient consent was not obtained because this study only uses patient’s information from the MyTB database.

## 3. Results

### 3.1. Sample Characteristics

Between 2014 and 2018, there were 24,277 TB patients registered in MyTB from Sabah. Of these, 1794 were identified as TB death, and 2000 controls were randomly selected from 20,557 TB patients who completed anti-TB treatment. A total of 1926 from the cohort were excluded, i.e., loss to follow-up (1533), changed diagnosis (181), failed treatment (36), not evaluated (17), and incomplete information of causes of TB death (159) (Figure 1). Of the 1794 TB deaths, 668 (37.2%) categorized as TB-related and 1126 (62.8%) as non-TB-related deaths. Generally, TB patients who died during treatment were older with a mean age of 52.0 years (sd ± 19.8) compared with the controls with a mean age of 39.0 years (SD ± 17.6). Sociodemographic, comorbid, disease, and treatment characteristics of cases and controls are presented in Table 1.

### 3.2. Trend of TB Death

The TB death rate increased from 9.0/100,000 population in 2014 to 11.4/100,000 population in 2018 while the TB notification rate showed a small reduction over the same period. The proportion of TB-related deaths from total TB deaths remained around 37% each year during the study period (Table 2). All-cause TB deaths were mainly not directly related to the TB disease itself.

### 3.3. Timing of TB Death

Of the 1794 TB deaths, 142 (7.9%) occurred before anti-TB treatment could be started. These deaths happened soon after TB diagnosis was made or during TB work-up before TB diagnosis was confirmed. Most (74.9%) of all TB deaths occurred in the first two months of TB diagnosis (Figure 2). The median time to death for all patients who died during anti-TB treatment was 21.5 days (IQR 5.0–61.0). The median time to death for patients who died of TB-related and non-TB-related causes was 7.0 days (IQR 2.0–30.0) and 34.0 days (IQR 11.0–81.0), respectively. Being a non-citizen (aOR = 2.32, 95% CI 1.44–3.72) and having far-advanced lung lesion on chest radiography (CXR) (aOR = 3.43, 95% CI 2.02–5.84) increased the odds of dying before anti-TB treatment started. The risk factors for TB death before treatment started are shown in Table 3.

### 3.4. Causes of TB Death

Sepsis is both responsible for TB-related (22.3%) and non-TB-related (17.9%) death. In TB-related death, sepsis occurred because of underlying advanced TB or disseminated TB. In contrast, in non-TB-related death, it was primarily because of the patient also having multiple comorbidities. Advanced TB or having disseminated TB at presentation is the main cause of TB-related death, accounting for 62.9% of all causes of TB-related death. Pneumonia, existing chronic lung disease, cancer, AIDS, cardiac events, and other comorbidities are common specific causes of non-TB-related death. The details on specific causes of TB-related and non-TB-related death are shown in Table 4.

### 3.5. Risk Factors of TB Death

In multivariable binary logistic regression analysis, in which all-cause TB deaths were compared to the controls, older age, male, socioeconomically disadvantaged, human immunodeficiency virus (HIV) positive, and advanced TB disease were independent risk factors for all-cause TB death. Patients on anti-TB treatment with FDC and with their treatment supervision under DOT achieved, are less likely to die from all-cause TB death. The details of risk factors for all-cause TB death are shown in Table 5.

In multinominal logistic regression analysis in which TB-related and non-TB-related deaths were compared to the controls, no formal education, smoking, and passive case detection were found to be independent risk factors for TB-related deaths but not for non-TB-related deaths. Non-citizen was found to be an independent risk factor for non-TB-related death but not for TB-related death. Older age was found to be an independent risk factor for both TB-related and non-TB-related deaths, but the effect is larger in non-TB-related deaths. Those aged 65 years and above experienced increased odds (aOR = 14.86, 95% CI 7.55–29.27) of dying compared to those aged 14 years and below from non-TB-related causes. However, the same age group experienced less odds of dying (aOR = 3.02, 95% CI 1.62–5.65) from TB-related causes. The other risk factors for both TB-related and non-TB related deaths are shown in Table 6. 

## 4. Discussion

Our analysis of TB surveillance data collected over five years from 2014 to 2018 showed an increasing trend of case fatality rate each year. The case fatality rate had increased from 6.6% in 2014 to 8.5% in 2018. The increasing fatality rate is not in line with the WHO End TB strategy milestone target to reduce TB deaths by 35% in 2020 compared to 2015. A majority (62.8%) of all TB deaths were due to non-TB-related causes, consistent with a study conducted among hospitalized TB patients in Taiwan [7]. Generally, TB deaths mainly involved older male patients who came from a low socioeconomic background.

Starting anti-TB treatment immediately after TB diagnosis may improve treatment outcomes. However, this study found that 7.9% of all TB deaths occurred before treatment started, occurring during TB work-up, or soon after TB diagnosis was made. In some instances, treatment cannot be started immediately at the time of diagnosis. It usually takes a few days for the patients to come back to the clinic after their TB work-up confirms their diagnosis. TB deaths before treatment started were associated with non-citizen and those with advanced lung lesions from their initial CXR. Advanced lung lesions at presentation indicate a delay in seeking treatment. As non-citizens contribute about 30% of all TB deaths, special attention is needed for this sub-group population. To detect TB early among this population, active case detection needs to be considered.

In this study, the first two months of treatment is a critical time as a majority of all-cause mortality among TB patients (74.9%) occurred during this period. This finding is similar to studies conducted in Ethiopia [8] and South Korea [9]. To address the high risk of death during the intensive phase of treatment, close monitoring and effective clinical management of TB patients need to be emphasized. In this study, the median time to death for all TB deaths of 21.5 days (IQR 5.0–61.0) is slightly shorter duration compared with 23 days (IQR 8–48) among TB patients in Taiwan [7]. The median time to death for patients who died of TB directly (7.0 days IQR: 2.0–30.0) was shorter than patients who died of non-TB-related causes (34.0 days IQR: 11.0–81.0). These findings are similar to the median survival time among TB patients in Taiwan [7] and the United States of America [10]. TB-related deaths were mainly due to advanced or disseminated TB at presentation, whereas non-TB-related deaths were due to existing comorbidities and sepsis.

In the multivariable binary logistic analysis, being male and older age were significantly associated with all-cause TB deaths, consistent with findings reported in Taiwan [11] and Nigeria [12]. Those staying in an urban area were at higher odds of dying while on anti-TB treatment. This observation could be due to the presence of a large migrant population in major towns in Sabah. Many of them with undocumented status face difficulties accessing standard healthcare, thus resulting in delaying their TB diagnosis and treatment. Studies elsewhere have also reported that marginalized populations often living in poverty experience a higher rate of TB mortality [12,13,14]. Disease characteristics such as retreatment cases, extra pulmonary tuberculosis, and advanced lesion on CXR were found to be independent risk factors for all-cause TB deaths. These TB disease characteristics at diagnosis indicate that patients seek healthcare late. Special TB control activities are needed for these marginalized populations to improve TB case detection and subsequently reduce TB deaths. Besides starting anti-TB treatment as early as possible, the choice of delivery and type of drugs given need special consideration to improve TB treatment outcomes. In this study, we found that patients on the fixed-dose combination (FDC) compared to loose tablets of anti-TB, and when DOT supervision was achieved were associated with reduced odds of dying. However, patients with TB–HIV co-infection were six times higher odds of dying compared to those who were HIV negative. Strengthening TB/HIV control program integration is crucial to improve case management of TB–HIV co-infected patients. Studies conducted elsewhere that showed favorable treatment outcomes of between 79.8% [5] and 89.8% [15] can be achieved with the targeted clinical management of TB–HIV co-infected patients, specifically for those with other risks.

The multinominal logistic regression analysis found similarities and differences in factors associated with TB-related and non-TB-related deaths. Older age is significantly associated with TB-related and non-TB-related deaths, but the effect was higher in non-TB-related deaths. This finding suggests that TB patients who died of non-TB-related causes are much older with multiple comorbidities compared with much younger TB patients who died of TB-related causes. Non-citizens had a 31% increased odds of dying from TB-related causes but not from non-TB-related causes. The high odds of dying from TB-related causes could be due to non-citizens presenting with more advanced disease. We also found that TB diagnosed through passive case detection was a significantly independent risk factor for TB-related death but not for non-TB-related death. Thus, to reduce deaths due to TB-related causes, more emphasis should be given to active case findings in addition to existing passive case findings to ensure TB diagnosis can be made in the early stage of the disease.

Some limitations in our study include the possibility of misclassification of the causes of TB deaths into either TB-related or non-TB-related. However, this misclassification is minimized as all included TB deaths in this study had been audited by a special committee at the district and state level. The actual number of TB deaths could be underestimated as we excluded deaths without information on the specific cause of death and those lost to follow-up. The strength of our study is that we analyze state-wide population-based surveillance data; thus, the findings can be generalized to the Sabah population and other states in Malaysia with similar population characteristics and TB burden. We also found important factors associated with TB deaths that could be used to develop a predictive model for TB mortality.

## 5. Conclusions

The yearly TB notification rate was relatively constant, but the TB mortality rate increased each year between 2014 and 2018. The majority of TB deaths occurred in the first two months of treatment and were attributed equally to TB-related and non-TB-related causes. TB-related deaths were primarily due to advanced disease or disseminated TB, whereas non-TB-related deaths were primarily due to existing comorbidities and sepsis resulting in septic shock. Many important independent risk factors for TB deaths were identified, which are useful to re-strategize TB control programs to address the increasing TB mortality rate. These factors can also be used to develop a predictive model for TB mortality. A predictive model is useful to identify TB patients at higher risk of mortality early so that specific interventions can be rendered accordingly.

## Figures and Tables

**Figure 1 ijerph-18-09740-f001:**
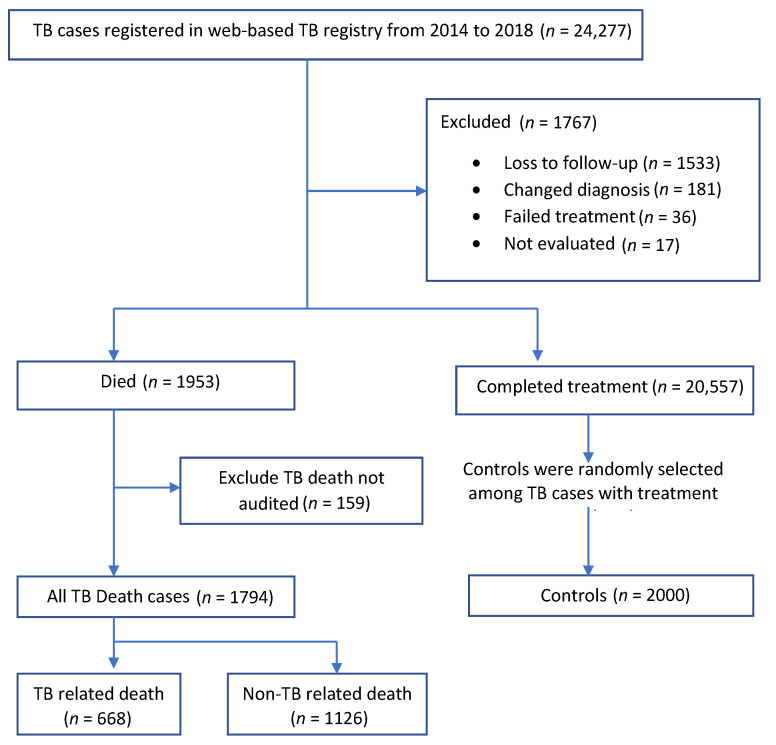
Selection of cases and controls from TB registry 2014–2018.

**Figure 2 ijerph-18-09740-f002:**
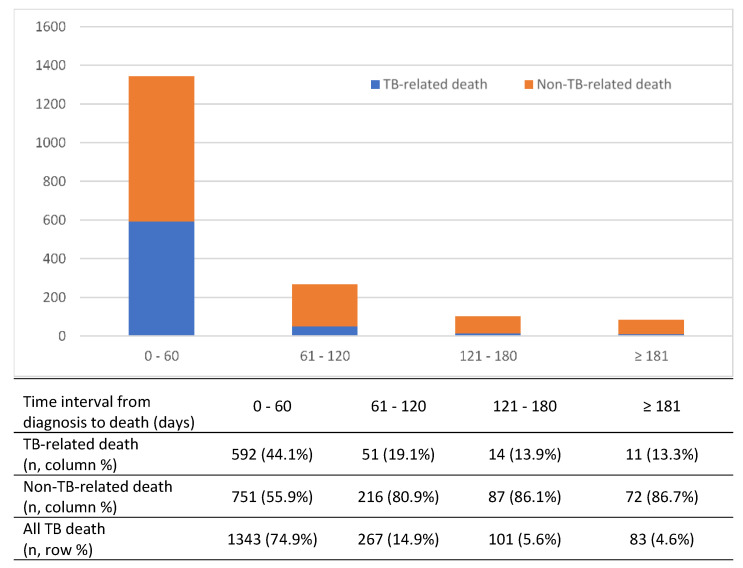
Frequency of TB-related, non-TB-related, and all TB deaths by timing of death.

**Table 1 ijerph-18-09740-t001:** Characteristics of cases and controls.

Variables	TB Death			Control *n* (%)	*p*-Value
	TB-Related *n* (%)	Non-TB Related *n* (%)	All TB Death *n* (%)		
Sociodemographic Characteristics					
Age (mean ± SD), years	44.6 ± 20.8	56.4 ± 17.8	52.0 ± 19.8	39.0 ± 17.6	<0.001 **
Age groups					
0–14	34 (5.1)	15 (1.3)	49 (2.7)	78 (3.9)	<0.001
15–44	303 (45.4)	250 (22.2)	553 (30.8)	1156 (57.8)	
45–64	202 (30.2)	460 (40.9)	662 (36.9)	585 (29.3)	
≥65	129 (19.3)	401 (35.6)	530 (29.5)	181 (9.1)	
Gender					
Female	265 (39.7)	353 (31.3)	618 (34.4)	817 (40.9)	<0.001
Male	403 (60.3)	773 (68.7)	1176 (65.6)	1183 (59.2)	
Place of residence					
Rural	462 (70.0)	778 (69.6)	1240 (69.7)	1462 (73.1)	0.010
Urban	198 (30.0)	340 (30.4)	538 (30.3)	527 (26.5)	
Nationality					
Malaysian	379 (56.7)	937 (83.2)	1316 (73.4)	1484 (74.2)	0.555
Non-Malaysian	289 (43.3)	189 (16.8)	478 (26.6)	516 (25.8)	
Education level					
Tertiary	8 (1.2)	16 (1.4)	24 (1.3)	49 (2.5)	<0.001
Secondary	136 (20.4)	350 (31.1)	486 (27.1)	841 (42.1)	
Primary	128 (19.2)	291 (25.8)	419 (23.4)	462 (23.1)	
No formal education	396 (59.3)	469 (41.7)	865 (48.2)	648 (32.4)	
Income regular					
No	504 (75.4)	780 (69.3)	1284 (71.6)	1182 (59.1)	<0.001
Yes	164 (24.6)	346 (30.7)	510 (28.4)	818 (40.9)	
Comorbid characteristics					
Smoking status					
No	497 (74.4)	777 (69.0)	1274 (71.0)	1405 (70.3)	0.606
Yes	171 (25.6)	349 (31.0)	520 (29.0)	595 (29.8)	
Diabetes mellitus status					
No	620 (92.8)	953 (84.6)	1573 (87.7)	1834 (91.7)	<0.001
Yes	48 (7.2)	173 (15.4)	221 (12.3)	166 (8.3)	
HIV status					
Negative	643 (97.4)	1022 (92.2)	1665 (94.1)	1967 (98.4)	<0.001
Positive	17 (2.6)	87 (7.8)	104 (5.9)	32 (1.6)	
Disease characteristics					
Category of TB					
New case	613 (91.8)	1003 (89.1)	1616 (90.1)	1884 (94.2)	<0.001
Retreatment case	55 (8.2)	123 (10.9)	178 (9.9)	116 (5.8)	
Type of case detection					
Passive	625 (93.6)	1015 (90.1)	1640 (91.4)	1801 (90.1)	0.148
Active	39 (5.8)	111 (9.9)	154 (8.6)	199 (10.0)	
BCG scar					
No	363 (54.3)	482 (42.8)	845 (47.1)	604 (30.2)	<0.001
Yes	305 (45.7)	644 (57.2)	949 (52.9)	1396 (69.8)	
TB anatomical site					
PTB	602 (90.1)	932 (82.8)	1534 (85.5)	1779 (89.0)	0.001
Extra PTB	66 (9.9)	194 (17.2)	260 (14.5)	221 (11.1)	
Initial sputum AFB					
Positive	541 (82.7)	681 (63.1)	1222 (70.5)	1434 (71.7)	0.032
Negative	113 (17.3)	398 (36.9)	511 (29.5)	512 (25.6)	
CXR status at diagnosis					
Minimal/normal	181 (28.1)	510 (46.8)	691 (39.9)	1103 (57.1)	<0.001
Moderate	336 (52.2)	474 (43.5)	810 (46.7)	741 (38.4)	
Advanced	127 (19.7)	105 (9.6)	232 (13.4)	87 (4.5)	
MDRTB status					
No	480 (99.6)	839 (99.9)	1319 (99.8)	1691 (100.0)	0.084 *
Yes	2 (0.4)	1 (0.1)	3 (0.2)	0	
TB meningitis					
No	67 (58.8)	205 (77.9)	272 (72.1)	255 (93.1)	<0.001
Yes	47 (41.2)	58 (22.1)	105 (27.9)	19 (6.9)	
TB miliary					
No	93 (81.6)	232 (88.2)	325 (86.2)	268 (97.8)	<0.001
Yes	21 (18.4)	31 (11.8)	52 (13.8)	6 (2.2)	
Treatment characteristics					
Treatment regime (intensive phase)					
FDC	341 (51.0)	563 (50.0)	904 (50.4)	1428 (71.4)	<0.001
Loose tablet	327 (49.0)	563 (50.0)	890 (49.6)	572 (28.6)	
Treatment delay (≥14 days)					
No	659 (98.7)	1105 (98.1)	1764 (98.3)	1975 (98.8)	0.277
Yes	9 (1.3)	21 (1.9)	30 (1.7)	25 (1.3)	
DOT achieved (intensive phase)					
No	268 (41.4)	329 (29.8)	597 (34.1)	7 (0.4)	<0.001
Yes	380 (58.6)	774 (70.2)	1154 (65.9)	1993 (99.7)	

*p*-value to compare all TB death cases with controls using Chi-square test. * Fisher’s exact test, ** Independent *t*-test; BCG = Bacillus Calmette–Guérin; CXR = Chest radiograph; FDC = Fixed-dose combination; DOT = Directly observed therapy.

**Table 2 ijerph-18-09740-t002:** Trend of TB death, 2014–2018.

Year	2014	2015	2016	2017	2018	Total
Total TB cases registered	4747	4464	4953	5105	5008	24,277
Total TB death cases	312	327	415	473	426	1953
Non-TB-related death	198	213	255	241	219	1126
TB-related death	114	110	148	168	128	668
TB-related death (%)	36.5%	34.1%	36.7%	41.1%	36.9%	37.2%
TB notification rate (per 100,000 population)	136.2	126.0	137.5	139.4	134.6	134.7 *
TB death rate (per 100,000 population) **	9.0	9.2	11.5	12.9	11.4	10.8 *
Case fatality rate (%)	6.6%	7.3%	8.4%	9.3%	8.5%	8.0%

* Average. ** Chi-square for trend = 21.83, *p* < 0.001.

**Table 3 ijerph-18-09740-t003:** Binary logistic regression analysis for risk factors of all-cause TB death before treatment started.

Variables	Time of TB Death		Univariable Analysis	Multivariable Analysis
	After Treatment Started, *n* (%)	Before Treatment Started, *n* (%)	cOR (95% CI)	aOR (95% CI)
Age (mean ± SD), years	45.2 ± 19.8	44.1 ± 19.8	0.99 (0.98–1.01)	
Age groups				
0–14	43 (87.8)	6 (12.2)	1	1
15–44	487 (88.1)	66 (11.9)	0.95 (0.39–2.37)	0.71 (0.27–1.84)
45–64	618 (93.4)	44 (6.6)	0.51 (0.21–1.26)	0.83 (0.31–2.21)
≥65	504 (95.1)	26 (4.9)	0.37 (0.14–0.95)	0.87 (0.30–2.48)
Gender				
Female	562 (90.9)	56 (9.1)	1	1
Male	1090 (92.7)	86 (7.3)	0.79 (0.56–1.13)	1.01 (0.68–1.50)
Place of residence				
Rural	1147 (92.5)	93 (7.5)	1	1
Urban	493 (91.6)	45 (8.4)	1.13 (0.78–1.63)	1.19 (0.79–1.78)
Nationality				
Malaysian	1242 (94.4)	74 (5.6)	1	1
Non-Malaysian	410 (85.8)	68 (14.2)	2.78 (1.97–3.94)	2.32 (1.44–3.72)
Diabetes mellitus status				
No	1443 (91.7)	130 (8.3)	1	1
Yes	209 (94.6)	12 (5.4)	0.64 (0.35–1.17)	0.76 (0.36–1.62)
HIV status				
Negative	1543 (92.7)	122 (7.3)	1	1
Positive	93 (89.4)	11 (10.6)	1.50 (0.78–2.87)	3.66 (1.80–7.43)
Type of case detection				
Passive	1507 (91.9)	133 (8.1)	1	1
Active	145 (94.2)	9 (5.8)	0.70 (0.35–1.41)	0.36 (0.13–0.99)
BCG scar				
No	765 (90.5)	80 (9.5)	1	1
Yes	887 (93.5)	62 (6.5)	0.67 (0.47–0.94)	0.78 (0.49–1.23)
CXR status at diagnosis				
Minimal/normal	650 (94.1)	41 (5.9)	1	1
Moderately advanced	753 (93.0)	57 (7.0)	1.20 (0.79–1.82)	1.53 (0.99–2.36)
Far-advanced	203 (87.5)	29 (12.5)	2.27 (1.37–3.74)	3.43 (2.02–5.84)

**Table 4 ijerph-18-09740-t004:** Specific causes of TB-related and non-TB-related death.

Cause of TB Death	Frequency	%
TB-related death		
Advanced/disseminated	420	62.9%
Sepsis	149	22.3%
TB meningitis	42	6.3%
TB miliary	10	1.5%
Acute respiratory distress syndrome	10	1.5%
Underlying HIV	17	2.5%
Severe pneumonia	6	0.9%
MDRTB	2	0.3%
Others	12	1.8%
Sub-total	668	100.0%
Non-TB-related death		
Sepsis	201	17.9%
Pneumonia-HAP	117	10.4%
Pneumonia-CAP	48	4.3%
Pneumonia-other	95	8.4%
Lung-other	52	4.6%
Lung-COAD	20	1.8%
Carcinoma-lung	48	4.3%
Carcinoma-other	93	8.3%
AIDS	94	8.3%
Heart-acute coronary syndrome	91	8.1%
Heart-congestive heart failure	23	2.0%
Heart-other	9	0.8%
Stroke	29	2.6%
Meningitis-bacterial, other	20	1.8%
Upper gastrointestinal bleeding	25	2.2%
Liver failure–drug induced	15	1.3%
End stage renal failure	15	1.3%
Injury/accident	9	0.8%
Other	41	3.6%
Indetermined	81	7.2%
Subtotal	1126	100.0%

**Table 5 ijerph-18-09740-t005:** Binary logistic regression analysis of risk factors for all-cause TB death.

Variables	Univariable Analysis	Multivariable Analysis
	cOR (95% CI)	aOR (95% CI)
Age groups		
0–14	1	1
15–44	0.76 (0.53–1.10)	0.97 (0.58–1.62)
45–64	1.80 (1.24–2.62)	2.61 (1.56–4.35)
≥65	4.66 (3.14–6.92)	7.46 (4.39–12.67)
Gender		
Female	1	1
Male	1.31 (1.15–1.50)	1.17 (0.97–1.38)
Place of residence		
Rural	1	1
Urban	1.20 (1.04–1.39)	1.47 (1.22–1.77)
Income regular		
No	1.74 (1.52–1.99)	1.46 (1.22–1.77)
Yes	1	1
HIV status		
Negative	1	1
Positive	3.84 (2.57–5.74)	5.61 (3.51–8.97)
Category of TB		
New case	1	1
Retreatment case	1.79 (1.40–2.28)	1.63 (1.21–2.21)
TB anatomical site		
PTB	1	1
Extra PTB	1.36 (1.13–1.65)	1.86 (1.43–2.42)
CXR status at diagnosis		
Minimal/normal	1	1
Moderately advanced	1.75 (1.52–2.00)	1.71 (1.43–2.05)
Far-advanced	4.26 (3.27–5.54)	4.56 (3.31–6.28)
Treatment regime (intensive phase)		
FDC	1	1
Loose tablet	2.46 (2.15–2.81)	2.11 (1.78–2.52)
DOT (intensive phase)		
No	1	1
Yes	0.007 (0.003–0.014)	0.007 (0.003–0.014)

Hosmer–Lemeshow test, *p* = 0.156; Nagelkerke pseudo R^2^ = 0.481.

**Table 6 ijerph-18-09740-t006:** Multinominal logistic regression analysis for association between sociodemographic, disease, and treatment characteristics with TB-related and non-TB-related death.

Variables	TB-Related Death		Non-TB Related Death	
	Univariable Analysis	Multivariable Analysis	Univariable Analysis	Multivariable Analysis
	cOR (95% CI)	aOR (95% CI)	cOR (95% CI)	aOR (95% CI)
Age groups				
0–14	1	1	1	1
15–44	0.60 (0.39–0.92)	0.87 (0.48–1.56)	1.13 (0.64–1.99)	1.53 (0.79–2.97)
45–64	0.79 (0.51–1.22)	1.31 (0.72–2.38)	4.09 (2.32–7.20)	5.26 (2.71–10.18)
≥65	1.64 (1.03–2.59)	3.18 (1.70–5.94)	11.52 (6.45–20.57)	14.89 (7.57–29.32)
Nationality				
Citizen	1	1	1	1
Non-citizen	2.19 (1.83–2.63)	1.86 (1.46–2.37)	0.58 (0.48–0.70)	0.71 (0.56–0.89)
Place of residence				
Rural	1	1	1	1
Urban	1.19 (0.98–1.44)	1.54 (1.21–1.94)	1.21 (1.03–1.43)	1.47 (1.20–1.79)
Income regular				
No	2.13 (1.75–2.59)	1.63 (1.28–2.07)	1.56 (1.34–1.82)	1.32 (1.08–1.60)
Yes	1	1	1	1
Smoking				
No	1	1	1	1
Yes	0.81 (0.67–0.99)	0.73 (0.57–0.93)	1.06 (0.91–1.24)	0.96 (0.78–1.17)
Category of TB				
New case	1	1	1	1
Retreatment case	1.46 (1.04–2.03)	1.59 (1.08–2.36)	1.99 (1.53–2.59)	1.80 (1.31–2.47)
Type of case detection				
Active	1	1	1	1
Passive	1.61 (1.14–2.26)	1.57 (1.02–2.40)	0.99 (0.78–1.26)	1.02 (0.75–1.40)
CXR status at diagnosis				
Minimal/normal	1	1	1	1
Moderately advanced	2.76 (2.26–3.39)	2.27 (1.79–2.88)	1.38 (1.18–1.62)	1.23 (1.02–1.49)
Far-advanced	8.89 (6.49–12.19)	6.58 (4.56–9.50)	2.61 (1.93–3.53)	2.47 (1.74–3.53)
Treatment regime (intensive phase)				
FDC	1	1	1	1
Loose tablet	2.39 (2.00–2.87)	2.15 (1.72–2.68)	2.49 (2.15–2.91)	2.41 (2.00–2.90)
DOT (intensive phase)				
No	1	1	1	1
Yes	0.005 (0.002–0.011)	0.006 (0.003–0.013)	0.008 (0.004–0.018)	0.008 (0.004–0.018)

Pearson Chi-square goodness of fit, *p* = 0.364; Nagelkerke pseudo R^2^ = 0.458.

## Data Availability

The data used to produce the findings of this study are available from the author, R.A., but will not be made available publicly as the authors have no permission to do so from the Ministry of Health Malaysia.

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
