# Peer review of "Tuberculosis Death Epidemiology and Its Associated Risk Factors in Sabah, Malaysia"

_ijerph, 2021, doi:10.3390/ijerph18189740_

Round 1

Reviewer 1 Report

Review of Tuberculosis death epidemiology and its associated risk factors

This is a good paper on risk factors for TB mortality using committee validated TB case data from public hospitals and clinics. This paper will be of possible interest to those working in the field of TB, and those interested in risk factors for TB mortality in a sub-national area of Malaysia.

I have no objection to seeing this research published, but I do wonder if IJERPH is the most appropriate journal. While the writers do a good job covering TB in this area of the world, there is no environmental component to the paper. I do not believe that the paper addresses “the impacts of natural phenomena and anthropogenic factors on the quality of our environment, the interrelationships between environmental health and the quality of life, as well as the socio-cultural, political, economic, and legal considerations related to environmental stewardship, environmental medicine, and public health. “ Thus, I would recommend that the editors of IJERPH with the authors to find a better suited journal.  

That being said, I make the following suggestions:

Abstract:

  1. Please present the national TB notification rate and TB CFR.
  2. You mention and increase from 9.0 to 11.4 cases per 100,000 from 2014 to 2018. Was this change statistically significant?
  3. “TB-related deaths were primarily due to advanced 18 disease or disseminated TB, whereas non-TB-related deaths were primarily due to existing comorbidities.”

Introduction:

  1. Lines 25-6 Please mention the scientific name of the pathogen.
  2. Line 25-6 Please cite this information.

Methods

  1. Please harmonize your use of the word “cases.” It is often hard to distinguish a TB “case” from a “case” in your study (dead people.) This is a major problem in this section and should be corrected.
  2. MAJOR ISSUE: “Controls were randomly selected from TB cases 63 with completed treatment in the same period as cases.” I’m confused. I thought that “controls” were all TB positive persons who had completed treatment. If I am understanding this correctly, you have defined a “period” (it is unclear what the “period” is, day? Month? Year? Define this.) and then you match each TB death in that period to a control? Or are you just using this as a denominator? Is it really hard to understand what you strategy is here. Please clarify this. This is a problem that MUST be dealt with before publication. I am sure that your strategy is acceptable but please clarify what you did so the reader can understand. Honestly, I am completely unclear on what you did here.
  3. Section 2.2.1: Please explain your motivation in including non-TB deaths. Why include them?
  4. Section 2.2.2: I have no comments here, though I wonder how reliable DOT is in this context. Some discussion of data reliability would be helpful here as many readers might not be familiar with health systems in this part of the world.
  5. Also, was there missing data? If so, what did you do about it?
  6. Section 2.3: “A simultaneous forced-entry method was employed in multivariable analysis.” Again, I am unclear here. What variables were included in the multivariate model? Was there any attempt made to select an optimal model? Including large numbers of variables in a regression model is often not informative, given the possibility of high multi-collinearity among the variables. Thus, you might consider a “best model” that include variables that can be used to describe TB mortality in an informative way. For example, you could try using a backward (or forward) selection procedure and then selectively remove non significant variables until you reach the model which has the lowest AIC value. While this model may not have all of the variables you used, it may have the most important set of variables and might provide useful insights. Please clarify what your model building procedure was here, and what you hoped to achieve with it.
  7. “Multinominal logistic regression analysis was used for comparison between cases of 119 TB-related death and non-TB-related death with the controls to identify the factors asso- 120 ciated with TB-related and non-TB-related deaths.” Again, please clarify your goals here. I am unclear as to why you included non-TB related deaths.

Results

  1. I appreciate Figure 1. It is very helpful. But it also leads me to wonder exactly how controls were selected. Please clarify the strategy in the methods. Why did you only select 2,000 controls? You had 20,557 people to select from. I would suggest having a control to case ratio of at least 2:1.
  2. I realize you might not know, but what are typical reason for loss to follow up? Do your excluded participants introduce any level of bias into the analysis?
  3. Again, I am very confused about how the controls were selected. Were you matching cases and controls based on year? And, again, why so few when you had so many to choose from?
  4. Table 1: This is good, but I am confused by the p-value. What does it actually mean? And why have “Tb death” and then “non TB death” under it. It would seem that’s not actually TB death. Perhaps this is just a problem of semantics, but I would leave Non TB related death out of this table. I don’t understand what it adds.
  5. I would like to see a comparison of risk factors for TB death vs those who completed treatment (did not die). Thus, the p-value should represent the p value from whatever statistical test you used to compare those two. (I realize that you test for risk factors in Table 3.)
  6. “p-Value to compare all TB death cases with controls using Chi-square test.” As it stands here, I don’t know what this pvalue represents because you haven’t been specific about what “TB death cases” means. I am still unconvinced that non-TB related death belongs here. I am not suggesting that it should not be here, but I think you need to do a better job of justifying it.
  7. Table 2: This is helpful to show the reader that there might be some evidence to suggest trend, but did you statistically test this? Please include statistical tests for trend.
  8. Figure 2: It seems like most TB related deaths occur quickly.
  9. Table 3: This is a nice table and a good segue from Figure 2, but I’m again confused by the model building process here. You have 10 variables in the model. Some are going to be highly correlated (education and urban/rural). Some aren’t going to make much sense together (age and education, a child of 5 has no education). I would suggest removing the education variable from the multivariate model, unless you restrict the model to only adult patients.
  10. I think your strongest conclusions from Table 3 are from your univariate analyses. BCG is protective, which we would expect. Far advanced and being non-Malaysian are predictive in the univariate results. Honestly, I don’t know what is to be gained from the multivariable analysis here since you don’t do any sort of model selection. I would actually suggest removing it and focusing on your univariate results.
  11. I’m assuming there is missing data here. Was there any attempt to account for missing data?
  12. Table 5: Again, this is nice, but I think you need to think about the variables used in the multivariable model. If you are going to include children, remove the education variable. If you want to keep the education variable, limit the data set to adults.
  13. I don’t’ really understand what the p value means here. Leave it out. You have confidence intervals for your odds ratios. You have some really interesting results. Focus on these results in your discussion.
  14. It is interesting that significance of all variables does not change in the multivariable model. There are all strong predictors.
  15. You say “all cause TB death” Did you try models for those who died of TB (and not non-TB related death)?

Author Response

Response to Reviewer 1 Comments

This is a good paper on risk factors for TB mortality using committee validated TB case data from public hospitals and clinics. This paper will be of possible interest to those working in the field of TB, and those interested in risk factors for TB mortality in a sub-national area of Malaysia.

I have no objection to seeing this research published, but I do wonder if IJERPH is the most appropriate journal. While the writers do a good job covering TB in this area of the world, there is no environmental component to the paper. I do not believe that the paper addresses “the impacts of natural phenomena and anthropogenic factors on the quality of our environment, the interrelationships between environmental health and the quality of life, as well as the socio-cultural, political, economic, and legal considerations related to environmental stewardship, environmental medicine, and public health. “Thus, I would recommend that the editors of IJERPH with the authors to find a better suited journal.  

That being said, I make the following suggestions:

Abstract:

  1. Please present the national TB notification rate and TB CFR.
  2. You mention and increase from 9.0 to 11.4 cases per 100,000 from 2014 to 2018. Was this change statistically significant?
  3. “TB-related deaths were primarily due to advanced 18 disease or disseminated TB, whereas non-TB-related deaths were primarily due to existing comorbidities.”

Response: 1. TB incidence rate was added. 

  1. The increase from 9.0 to 11.4 per 100 000 is statistically significant (chi square

        for trend=21.83, p <0.001). “significantly’’ was added in the abstract and the        

        information was also added as footnote of Table 2.

  1. Correction done in the abstract as per reviewer suggestion

Introduction:

  1. Lines 25-6 Please mention the scientific name of the pathogen.
  2. Line 25-6 Please cite this information.

 Response: correction done as per reviewer comments

Methods

  1. Please harmonize your use of the word “cases.” It is often hard to distinguish a TB “case” from a “case” in your study (dead people.) This is a major problem in this section and should be corrected.

Response: Thank you for pointing this out. We agree that the term “cases” created confusion as it can be refereed to TB cases or cases (TB death) as defined in this study. Correction was made in the text accordingly whereby “cases” in the text is referring to TB death.

  1. MAJOR ISSUE: “Controls were randomly selected from TB cases 63 with completed treatment in the same period as cases.” I’m confused. I thought that “controls” were all TB positive persons who had completed treatment. If I am understanding this correctly, you have defined a “period” (it is unclear what the “period” is, day? Month? Year? Define this.) and then you match each TB death in that period to a control? Or are you just using this as a denominator? Is it really hard to understand what you strategy is here. Please clarify this. This is a problem that MUST be dealt with before publication. I am sure that your strategy is acceptable but please clarify what you did so the reader can understand. Honestly, I am completely unclear on what you did here.

Response: Thank you for pointing this out. We agree the sentence needed to be rephrased to make it clear what was done. We rephrase the sentence as “Control were randomly selected from TB patients registered between 2014 and 2018 who completed anti-TB treatment”

  1. Section 2.2.1: Please explain your motivation in including non-TB deaths. Why include them?

Response: The TB control program in Malaysia, all TB death among TB patients will be audited by a special committee (as described in the manuscript) to determine the specific cause of death and the TB death will be further classified as TB-related death (directly due to the TB disease itself) or non-TB related death (whereby TB patients died unlikely due to the TB disease itself but most likely due to other conditions e.g comorbid or accidents etc which is not directly related to TB). Thus, we would like to see if there are any difference in its causes. Of course, the first and main focus will be to identify risk factors for all-cause TB death vs control (TB patients who completed anti-TB treatment)

  1. Section 2.2.2: I have no comments here, though I wonder how reliable DOT is in this context. Some discussion of data reliability would be helpful here as many readers might not be familiar with health systems in this part of the world.

Response: Additional information on DOT implementation was inserted in the manuscript.

  1. Also, was there missing data? If so, what did you do about it?

Response: The variables with many missing value e.g. TB meningitis, TB miliary, MDRTB status were not included in the logistic regression to determine risk factors. Most of variables included in logistic regression are complete and some with small number of missing value (less than 20) which we consider has negligible effect.

  1. Section 2.3: “A simultaneous forced-entry method was employed in multivariable analysis.” Again, I am unclear here. What variables were included in the multivariate model? Was there any attempt made to select an optimal model? Including large numbers of variables in a regression model is often not informative, given the possibility of high multi-collinearity among the variables. Thus, you might consider a “best model” that include variables that can be used to describe TB mortality in an informative way. For example, you could try using a backward (or forward) selection procedure and then selectively remove non significant variables until you reach the model which has the lowest AIC value. While this model may not have all of the variables you used, it may have the most important set of variables and might provide useful insights. Please clarify what your model building procedure was here, and what you hoped to achieve with it.

Response: We screen important variables statistically and the test result shown in table 1. Variables with a p value <0.05 consider as important to be included in multivariable analysis. However, some variables with p <0.05 but high number of missing value or high correlated with other variable were excluded in univariable and multivariable logistic regression. With the above screening, we only consider 11 variables (as in table 5) to be consider in logistic regression. We first perform a univariable logistic regression to estimate the OR with its 95%CI, and subsequently employed multivariable logistic regression just to get better estimate of OR and 95%CI after adjusting for the other variables in the model. Anyway, we did attempt using backward / forward method in multivariable logistic regression and the result is similar (able to determine the same significant variables). Thus, we would like to maintain our original method as mentioned in the manuscript.

  1. “Multinominal logistic regression analysis was used for comparison between cases of 119 TB-related death and non-TB-related death with the controls to identify the factors asso- 120 ciated with TB-related and non-TB-related deaths.” Again, please clarify your goals here. I am unclear as to why you included non-TB related deaths.

Response: As explained in response of question 3 above.

Results

  1. I appreciate Figure 1. It is very helpful. But it also leads me to wonder exactly how controls were selected. Please clarify the strategy in the methods. Why did you only select 2,000 controls? You had 20,557 people to select from. I would suggest having a control to case ratio of at least 2:1.

Response: As in the response question 1 (method section). We decided to do a 1:1 case control ratio as the number of cases (TB deaths) is large enough, which in this study the total sample size is 3794 (1794 cases: 2000 controls). We of the opinion that having 1:2 case to control ratio will give similar results.

  1. I realize you might not know, but what are typical reason for loss to follow up? Do your excluded participants introduce any level of bias into the analysis?

Response: loss to follow up TB patients were excluded as we did not know their final status of treatment outcome. We did discuss this limitation in the discussion section. Some of these loss-to-follow up TB patients, their treatment outcome could be death thus will underestimate the TB death rate calculated in this study.

  1. Again, I am very confused about how the controls were selected. Were you matching cases and controls based on year? And, again, why so few when you had so many to choose from?

Response: As explained in question 1 (method section) and question 1 (result section)

  1. Table 1: This is good, but I am confused by the p-value. What does it actually mean? And why have “Tb death” and then “non TB death” under it. It would seem that’s not actually TB death. Perhaps this is just a problem of semantics, but I would leave Non TB related death out of this table. I don’t understand what it adds.

Response: As explained in question 3 and 6 (methods section). Thus, we would like to maintain the p value in table 1.

  1. I would like to see a comparison of risk factors for TB death vs those who completed treatment (did not die). Thus, the p-value should represent the p value from whatever statistical test you used to compare those two. (I realize that you test for risk factors in Table 3.)

Response: We did the analysis for this concerned (infact this is the main objective of this analysis) and the result as shown in Table 5 in the manuscript.

  1. “p-Value to compare all TB death cases with controls using Chi-square test.” As it stands here, I don’t know what this pvalue represents because you haven’t been specific about what “TB death cases” means. I am still unconvinced that non-TB related death belongs here. I am not suggesting that it should not be here, but I think you need to do a better job of justifying it.

Response: As explained in question 3 and 6 (methods section) and question 4 (results section).

  1. Table 2: This is helpful to show the reader that there might be some evidence to suggest trend, but did you statistically test this? Please include statistical tests for trend.

Response: We tested the TB death rate trend from 2014 to 2018 using chi-square for trend and included the findings as footnote of Table 2.

  1. Figure 2: It seems like most TB related deaths occur quickly.

Response: Yes

  1. Table 3: This is a nice table and a good segue from Figure 2, but I’m again confused by the model building process here. You have 10 variables in the model. Some are going to be highly correlated (education and urban/rural). Some aren’t going to make much sense together (age and education, a child of 5 has no education). I would suggest removing the education variable from the multivariate model, unless you restrict the model to only adult patients.

Response: We have check correlation between education and urban/rural and found to be very weak (r= -0.091). We agree to the reviewer suggestion and remove education level from the analysis. Table 3 was updated without the education variable and p value

  1. I think your strongest conclusions from Table 3 are from your univariate analyses. BCG is protective, which we would expect. Far advanced and being non-Malaysian are predictive in the univariate results. Honestly, I don’t know what is to be gained from the multivariable analysis here since you don’t do any sort of model selection. I would actually suggest removing it and focusing on your univariate results.

Response: As explained in question 6 (methods section). The multivariable analysis is to get better estimate of the OR and 95%CI after adjusting for other variables in the model.

  1. I’m assuming there is missing data here. Was there any attempt to account for missing data?

Response: As explained to question 5 in methods section

  1. Table 5: Again, this is nice, but I think you need to think about the variables used in the multivariable model. If you are going to include children, remove the education variable. If you want to keep the education variable, limit the data set to adults.

Response: We agree to the reviewer suggestion and remove education level from the analysis.

  1. I don’t’ really understand what the p value means here. Leave it out. You have confidence intervals for your odds ratios. You have some really interesting results. Focus on these results in your discussion.

Response: Thank you for the suggestion. We agree to remove the p value as it does not add additional information to the 95%CI. The table 5 was updated without the p value

  1. It is interesting that significance of all variables does not change in the multivariable model. There are all strong predictors.

Response: Agree

  1. You say “all cause TB death” Did you try models for those who died of TB (and not non-TB related death)?

Response: As explained (question 3 in methods section) earlier “all cause TB death” is referring to TB patients who died anytime from TB diagnosis is made to before completion of anti-TB treatment irrespective of the specific cause of death. Thus, all cause TB death (include TB-related death and non-TB related death). Clinically it is not easy to classify TB death into (TB-related death or non-TB related death) as all of them died while on TB treatment. In this study the classification is done through audit by a committee as describe in the manuscript. However, the main focus will be to look for risk factors for all cause TB death. In Table 5 the model is to determine risk factors for all cause TB death.

Reviewer 2 Report

Strengths of the manuscript:

The manuscript gives a very succinct rationale for the need to focus on TB epidemiology; putting the topic into perspective by referring to the WHO goals on TB reduction and also citing relevant statistics of TB deaths worldwide.

Explanatory variables were clearly defined

The following conclusion  is  also a strength of the manuscript given the number of non-citizens in Malaysia “ As non-citizen contributes about 30% 221 of all TB deaths, special attention is needed to this sub-group of population.”

ITEMS THAT NEED ATTENTION IN THE MANUSCRIPT

In line 36 the authors mention “Sabah state” as though the reader has knowledge of what this state is. It could have been useful to say something even about the demography of the state. This is important as they go on to say that strategies should be “tailored to different settings or  states”

The title  “Tuberculosis death epidemiology and its associated risk factors” sounds  like text book chapter title as it is too generic and not tailored to a specific setting. Note also that the authors wrote: “this study aims to provide 54 a comprehensive report on TB deaths epidemiology and its associated factors specifically 55 for the state of Sabah”

Under data availability statement Please substitute “we” with “the authors”

A number of epidemiological studies have been carried out a on a similar topic. How is yours nvel? 

Author Response

Response to Reviewer 2 Comments

Strengths of the manuscript:

The manuscript gives a very succinct rationale for the need to focus on TB epidemiology; putting the topic into perspective by referring to the WHO goals on TB reduction and also citing relevant statistics of TB deaths worldwide.

Explanatory variables were clearly defined

The following conclusion  is  also a strength of the manuscript given the number of non-citizens in Malaysia “ As non-citizen contributes about 30% 221 of all TB deaths, special attention is needed to this sub-group of population.”

Response: Thank you for these positive observation

ITEMS THAT NEED ATTENTION IN THE MANUSCRIPT

In line 36 the authors mention “Sabah state” as though the reader has knowledge of what this state is. It could have been useful to say something even about the demography of the state. This is important as they go on to say that strategies should be “tailored to different settings or  states”

Response: We included brief demography information about the state of Sabah in the Introduction section of the manuscript

The title  “Tuberculosis death epidemiology and its associated risk factors” sounds  like text book chapter title as it is too generic and not tailored to a specific setting. Note also that the authors wrote: “this study aims to provide 54 a comprehensive report on TB deaths epidemiology and its associated factors specifically 55 for the state of Sabah”

Response: We revise the title to “Tuberculosis death epidemiology and its associated risk factors in Sabah, Malaysia”

Under data availability statement Please substitute “we” with “the authors”

Response: Correction done

A number of epidemiological studies have been carried out a on a similar topic. How is yours nvel? 

Response: The TB death epidemiology and its associated factors are unique to this type of setting and may be comparable with other countries with similar sociodemographic background i.e. high number of immigrant population (marginalised population).
